# Causes and consequences of fever in Amazonian pregnant women: A large retrospective study from French Guiana

**Najeh Hcini**[1,2]*, **Véronique Lambert**[1], **Olivier Picone**[3], **Jean-Francois Carod**[4], **Mérédith Mathieu**[1], **Romane Cousin**[1], **Ferroudja Akli**[1], **Gabriel Carles**[1], **Célia Basurko**[5], **Léo Pomar**[6,7], **Loïc Epelboin**[2,8], **Mathieu Nacher**[5]

**1** Department of Obstetrics and Gynaecology, West French Guiana Hospital Center, Saint-Laurent-du-Maroni, French Guiana, **2** INSERM CIC1424 Centre d'Investigation Clinique Antilles Guyane, Cayenne, French Guiana, **3** Service Gynécologie Obstétrique, Hôpital Louis Mourier, Hôpitaux Universitaires Paris Nord Val de Seine, Assistance Publique: Hôpitaux de Paris, Université Paris Diderot, Colombes CEDEX, France, **4** Department of Biology, West French Guiana Hospital Center, Saint-Laurent-du-Maroni, French Guiana, **5** Centre d'Investigation Clinique Antilles-Guyane, Epidémiologie Clinique, CIC INSERM 1424, DRISP, Centre hospitalier de Cayenne, Cayenne, French Guiana, **6** Ultrasound and Fetal Medicine, Department Woman-Mother-Child, Lausanne University Hospital and University of Lausanne, Lausanne, Switzerland, **7** School of Health Sciences (HESAV), University of Applied Sciences and Arts Western Switzerland, Lausanne, Switzerland, **8** Department of Infectious and Tropical Diseases Cayenne General Hospital French Guiana France

* hcininajeh@gmail.com

**Data Availability Statement:** All relevant data are included in the paper.

**Funding:** The author(s) received no specific funding for this work.

## Abstract

### Objective

The aim of this study was to describe different causes and consequences of fever during pregnancy in Western French Guiana and along the Maroni River.

### Study design

A retrospective single-center study including all patients with a history of documented fever $\geq 38°C$ during pregnancy at the West French Guiana Hospital for 9 years. Postpartum fever and nosocomial infections were excluded. We focused on medical history and on clinical and biological findings. Causes were characterized as confirmed or uncertain and then classified as preventable or non-preventable.

### Results

A total of 940 pregnant women who experienced at least one episode of fever were included and compared to 23,811 deliveries who occurred during the same period without documented fever. Among them, 43.7% (411/940) were in labor. About 3.7% (35/940) of febrile pregnant women had at least two episodes of fever, while 0.3% (3/940) had a coinfection at the time of diagnosis, resulting in a total of 978 febrile episodes. Among them, causes remained unknown or uncertain in 7.6% (75/978) and 0.9% (9/978) of cases, respectively. Among confirmed causes of fever throughout pregnancy (n = 483), the most common known cause was arbovirus infection (146/483, 30.2%), followed by urinary tract infection

**Competing interests:** The authors have declared that no competing interests exist.

(134/483, 27.7%), chickenpox (27/483, 5.6%), and gastrointestinal (14/483, 2.9%) and pulmonary infections (10/483, 2%). Mothers with fever had a higher risk of cesarean section (19.8% vs 15.5%, aOR 1.3 [95% CI 1.14–1.6], stillbirth (5.5% versus 1.9%, aOR 2.7 [95% CI 2–3.7]), and preterm delivery < 34 weeks of gestation (7.2% vs 4.7%, aOR 1.5 [95% CI 1.2–2].

## Conclusions

In the Amazon region, causes of fever are diverse and often associated with epidemic waves, notably arboviruses. This must be considered when exploring possible causes of fever during pregnancy in these localities, including fetal anomalies and/or fetal loss. Physicians should consider the epidemiological context and avoid generalizations. Given the impact of emergent agents such as arboviruses on pregnancy, particular attention must be paid to the epidemiological context. This study can also help clinicians when managing fever in pregnant travelers or in their partner after having visited exposed areas. In this context, fetal abnormalities and adverse obstetric outcomes should be explored accordingly.

## Author summary

While the Amazon basin grapples with numerous epidemic and endemic infections, remains a lack of comprehensive information concerning the extent and impact of these infections on maternal and neonatal health. In this study, we have established that arboviral infections and urinary tract infections are the primary causes of fever in pregnant women. Notably, we have identified a significant association between these infections and adverse pregnancy outcomes, including stillbirth and prematurity. To address maternal and fetal infectious morbidity, it is crucial to implement close clinical and biological monitoring, along with providing appropriate treatment for pregnant women experiencing fever. Additionally, given the ongoing emergence of zoonotic diseases and the influence of climate change, which may facilitate the transmission of additional infections, there is an urgent need for the implementation of rapid diagnostic tools and the development of adapted surveillance and treatment strategies.

## Introduction

Severe infections during pregnancy and the post-partum period remain one of the main causes of maternal and fetal morbidity and mortality [1]. In this context, fever is a very common warning signal. There are specific clinical features during pregnancy that vary according to the trimester of pregnancy: presence of sympathetic signs of pregnancy, immunological changes, changes in the urinary tract, as well as metabolic, respiratory, vascular, and cardiogenic changes. These modifications can increase the severity of many infections such as malaria [2] and dengue fever [3]. Given the current emergence of multiple infectious agents and climate change, this topic requires more attention especially in neglected areas.

While the Amazon region has witnessed various infectious epidemics and continues to face ongoing outbreaks, there is limited data on the causes and consequences of fever in pregnant women in the often-precarious populations of the Amazonian basin [4,5]. In these regions, exploration of fever is often limited by a lack of adequate diagnostic facilities. Worldwide,

there are limited guidelines about managing fever in pregnant women living in or visiting these areas. However, both in pregnant mothers and infants are particularly at risk of adverse outcomes for infections [6,7]. Scientific data support that maternal fever during pregnancy, regardless the pathogen, can have a negative effect on the health of the offspring [8,9] and may even increase the risk of autism spectrum disorders [10]. Most reports on the impact of fever during pregnancy have focused on industrialized countries, while little information is available on precarious populations for which a clear link between malnutrition and the impact of infections on maternal health has been established [2,11]. A large proportion of the over 40 million people estimated to be living in the Amazon basin, suffers from social deprivation and malnutrition, often living far from healthcare providers that tend to be underfunded and ill-equipped. In this context, French Guiana forms something of an exception. Although located in South America, it is a French territory and therefore part of the European Union with a universal healthcare system and a widespread availability of modern diagnostic tools. At the same time, it is characterized by a multiethnic population very similar to that of other Amazonian territories.

The aim of this study was to describe the causes, risk factors, and the maternal and neonatal adverse outcomes of fever during pregnancy in West French Guiana and along the Maroni River.

## Methods

### Ethics statement

In compliance with the EU regulation 2016/679 regarding the protection of personal data, a privacy impact assessment of the study was carried out according to the methodology described by the French data protection guidelines. According to French regulations, written consent is not required; however, patients who refuse the use of their medical data for research purposes have been omitted from the computerized database. This study was approved by the Saint Laurent Hospital Ethics Committee (CHOG decision, January 18, 2023).

### Study design

This retrospective single-center cohort study included all pregnant women who presented with fever at the Department of Obstetrics and Gynecology of the Centre Hospitalier de l'Ouest Guyanais (CHOG), (referral center of Western French Guiana) during the 9-year period from January 1, 2013, and December 31, 2021, and who delivered at the same center. Only women who experienced fever and delivered after 18 weeks of gestation were included in the study since the delivery records only encompass deliveries occurring after 18 weeks of gestation (i.e. pregnant women with an early or late miscarriage, before 18 weeks, are not included in the delivery register, and were not included in this study). In this study, documented pregnancies with an auricular temperature (infrared emission detection) $\geq 38°C$ were included.

Exclusion criteria were patients opposing the use of their medical data, unknown pregnancy outcome, loss of medical record.

The term of pregnancy was measured using obstetrical ultrasound and calculated according to the craniocaudal length at the first trimester. If patients first presented in their second or third trimester, we considered the date of their last period and fetal head circumference to determine the term of pregnancy. Co-infections were defined by the simultaneous detection of two or more infectious fever-causing agents. Women who had more than one pregnancy during the study period were counted as separate participants. Only clinical documented fever was considered. Causes of fever were characterized as either confirmed, uncertain, or unknown. Confirmed causes were diagnosed using standard clinical, biological, and imaging

approaches. Causes were considered uncertain if a diagnosis could not be established with sufficient certainty or if information was missing. Infections were considered preventable if recognized effective methods existed to combat the identified cause (e.g., vaccination).

## Diagnosis

We measured maternal temperature using auricular thermometers. In our maternity emergency department, fever during pregnancy was managed by performing a first-line anamnestic, clinical, and biological work-up including blood tests with hemogram, plasma C-reactive protein (CRP), *Influenza virus* PCR in the presence of flu-like syndrome in nasopharyngeal swabs, and bacteriological samples with urine culture (UC), treponematoses, dengue, Q fever, vaginal swab, blood cultures for *Listeria Monocytogene*s and coproparasitological examinations in case of diarrhea. Recently, since May 2020, we added RT-qPCR or antigen testing for SARS-COV-2 in nasopharyngeal swabs.

This protocol screens for the presence of common infections specific to our geographical region, such as syphilis, parvovirus, malaria, leptospirosis, Q fever, Chagas, Chikungunya (CHIKV), dengue virus, and Mayaro. In 2016, screening for Zika virus (ZIKV) was added, and in 2019, Tonate virus. All diagnoses of arboviruses were conducted by the Institute Pasteur of French Guiana, the national reference center for arboviruses. Women who without etiological diagnosis at initial work-up would receive further explorations.

For viable pregnancies, clinical and fetal monitoring was performed. In case of suspicion of rupture of the membranes, we tested for the presence of Placental alpha 1-microglobulin (PAMG-1) in vaginal fluid. In case of uterine contraction, a cervical assessment was performed. Antipyretic treatment with paracetamol was systematically administered.

The type of second-line examination depended on maternal symptoms and clinical findings. Biological testing included parvovirus B19, rickettsioses, leptospirosis, cytomegalovirus (CMV), Epstein-Barr virus, and medical imaging tests (abdominal ultrasound, venous doppler, thoracic imaging if indicated by symptoms). Hospitalization was not systematic but depended on both clinical presentation and social condition (living far from hospital, pregnancy follow-up, health insurance coverage, etc.).

The Western French Guiana maternity protocol for managing fever during pregnancy is summarized in Table 1.

## Defining the causes of fever

We performed exhaustive data collection using medical and biological records to try and assign a cause of fever. Infections were considered confirmed or suspected based local standards of definition. Medical history, clinical investigation, biological findings, placental histology, and autopsy examination were also considered.

Indication of blood cultures included fever $\geq 38.3$ or hypothermia; cultures were repeated if necessary.

Diagnosis criteria were:

**Chikungunya virus (CHIKV):** Detection of the viral RNA genome by real-time RT-PCR targeting the NSP1 gene of CHIKV, as described by Panning et al. [12] or seroconversion of IgM/IgG antibodies in the collected samples. Both IgM and IgG anti-CHIKV-specific antibodies were screened in sera using an in-house enzyme-linked immunosorbent assay (IgM antibody capture ELISA and ELISA, respectively) as described by Talarmin et al [13].

**Zika virus (ZIKV):** Pregnant patients were defined as ZIKV positive either through a positive RT-PCR result (RealStar Zika Kit, Altona Diagnostics GmbH, Hamburg, Germany) from blood and/or urine samples or by anti-ZIKV IgM antibody detection using an in-house

**Table 1. Fever evaluation during pregnancy at the West French Guiana Hospital.**

| Elements of fever evaluation | Details |
|---|---|
| **Anamnesis and clinical evaluation** | Medical and obstetric history, maternal medical condition, previous fever, history of malaria, recent travel, fever in social circle, current pregnancy information, physical examination |
| **First-line anamnestic, clinical, and biological work-up** | Blood tests with hemogram, plasma C-reactive protein (CRP), bacteriological samples with urine culture (UC), treponematoses, vaginal swab, hemocultures including *Listeria monocytogenes* if fever $\geq$ 38.3°C or chills.<br>toxoplasmosis, rubella virus if negative before,<br>Syphilis, HIV, molecular or antigen testing for SARS-COV-2 testing through nasopharyngeal swabs (since May 2020),<br>PCR influenza if flu-like syndrome in nasopharyngeal swabs,<br>molecular malaria detection, Coxiella burnetiid serology (Q fever), Chagas, parvovirus, arboviruses molecular or/and serologic detection (depends on the sampling date relative to the onset of symptoms).<br>In cases of febrile diarrhea, coproparasitological and bacteriological analysis of fecal samples, including testing for salmonella, shigella, or campylobacter; in cases of pneumonia, testing for Streptococcus pneumoniae, and legionella pneumophila. |
| **Obstetrical assessment** | In case of suspicion of rupture of membranes, testing for presence of PAMG-1(Placental alpha 1-microglobulin) in vaginal fluid; cervical assessment in case of uterine contraction<br>fetal monitoring in viable fetus. |
| **Second biological work-up** | - Parvovirus B19, rickettsioses, leptospirosis, Epstein-Barr virus (EBV), cytomegalovirus (CMV)<br>- Liver, kidney, and thyroid function<br>- Screening for autoimmune diseases according to clinical findings: antiphospholipid antibodies, antinuclear antibodies<br>- Medical imaging tests according to clinical orientation<br>- The research for tuberculosis was not systematically conducted but guided by the patient's history and clinical findings.<br>- Request for expert advice |
| **At delivery** | If necessary, cord sampling for infectious agents, bacteriological analyses of placenta, placental pathology if indicated. |

MAC-ELISA test (based on whole virus antigens obtained in cell culture and on hyperimmune ascitic fluid) [14].

**Dengue fever:** Specific dengue IgM antibodies identified by MAC using an in-house protocol described by Talarmin et al. [13], viral RNA detection via PCR, and a Platelia Dengue NS1 Ag capture ELISA (Bio-Rad laboratories, Marnes-La-Coquette, France).

**Yellow fever**: RT-PCR and serological tests.

All testing for arboviruses was performed using RT-qPCR and serology at the National Reference Center of the Pasteur Institute, French Guiana.

**SARS-CoV-2:** SARS-CoV-2: SARS-CoV-2 real-time reverse-transcriptase polymerase-chain-reaction (RT-qPCR) assays using Xpert Xpress SARS-CoV-2 (Cepheid, Sunnyvale, USA) on nasopharyngeal swabs, or (if testing capability was lacking) rapid antigen tests: PAN-BIO COVID-19 Ag RAPID TEST DEVICE (PanBIO BY ABBOTT, North Chicago, IL).

**Influenza:** Positive RT-PCR nasopharyngeal swabs using GeneXpert Xpress Flu/RSV (Cepheid, Sunnyvale, CA).

**Listeriosis:** Positive blood culture for Listeria monocytogenes on Bactec FX (Becton-Dickinson, San Jose, CA) identified by MALDI-TOF on MALDI Biotyper sirius IVD System (Bruker Daltonics GmbH & Co. KG Life Sciences, Bremen, Germany.

**Syphilis:** Screening for syphilis was performed using a treponemal serologic test (TT), specifically the Treponema pallidum particle agglutination (TPPA) method with Chemiluminescent Microparticle Immunoassay (CMIA) technology (Abbott ARCHITECT, Wiesbaden, Germany), which was confirmed by a positive immunoblot test (Virotech Diagnostics GmbH, Dietzenbach, Germany). In cases of a positive result, this test was followed by a rapid plasma reagin (RPR) assay (Bio-Rad Laboratories, Hercules, CA), a quantitative nontreponemal test used to assess treponema activity. After confirming the diagnosis, women underwent a comprehensive clinical evaluation, which included anamnesis, medical history, physical examination, and biological testing. Based on this assessment, they were assigned a clinical stage.

**Acute pyelonephritis and urinary tract infection:** Acute pyelonephritis was diagnosed based on clinical findings of fever ($\geq 38°C$), flank pain, and costovertebral angle tenderness, and on laboratory findings (confirmation by urine culture). All diagnosed women were evaluated by physicians and faculty before admission to the hospital and were treated with intravenous fluid and antimicrobial agents, adjusting depending on the bacteriological results.

**Intra-uterine infection:** This diagnosis was retained in case of maternal fever ($\geq 38°C$), in the absence of alternative causes and the presence of at least 2 of the following signs: fetal tachycardia$>160$ bpm for at least 10min, uterine pain of labor, purulent fluid from the cervical canal according to National College of French Gynecologists and Obstetricians (CNGOF) guidelines [15]. Intrapartum fever and intrauterine infection were managed differently in accordance with the same guidelines.

**Respiratory infection:** Presence of clinical symptoms and signs, radiological findings with biological testing using BioFire Respiratory Panel 2.1 plus multiplex PCR (mPCR) for bacteria and viruses (bioMérieux, Craponne, France) and/or urinary antigen tests for pneumococcus and legionella pneumophila using BinaxNOW Urinary Antigen Test Kit (ABBOTT, North Chicago, IL). Sputum samples, collected as triplicates, were decontaminated, pelleted by centrifugation, examined for acid-fast bacilli using fluorescence microscopy, and tested by GeneXpert Mycobacterium tuberculosis/rifampicin assay (Cepheid, Sunnyvale, California). Acid-fast bacilli smear-positive results were graded according to CDC guidelines.

**Leptospirosis:** DNA detection by PCR (Biosynex Ampliquick Leptospira, France) was conducted within 7 days of symptom onset, or on serology (IgM or MAT) if conducted more than 7 days after fever onset.

***Coxiella burnetii***: IgG and IgM antibodies to phase I and II of C. burnetii by indirect fluorescent antibody (IFA) assay (Focus Diagnostics, Cyprus, CA).

**Malaria:** Alethia Isothermal loop amplification (LAMP) by Alethia Meridian Diagnostics & Cincinnati, OH), confirmation of positive samples was done by PALUTOP +4 OPTIMA rapid diagnostic test (Biosynex SA, Illkirch-Graffenstaden, France), and examination of thick and thin blood smears.

**Hepatitis A:** Serological testing, specifically the presence of IgM antibodies (Abbott, Chicago, USA) indicating acute hepatitis A infection.

**Fecal coproparasitological and bacteriological tests:** including testing for pathogens such as Salmonella sp., Shigella sp., pathogenic E. coli, Yersinia, and Campylobacter sp. in cases of febrile diarrhea. Stool culture used specific agar plates Biomérieux (Craponne, France). All grown colonies were identified using MALDI Biotyper sirius IVD System (Bruker Daltonics GmbH & Co. KG Life Sciences, Bremen, Germany) or with Vitek galleries by Biomérieux (Craponne, France). Antibiotic Susceptibility Testing was performed when needed.

**Blood culture:** Positive blood culture on Bactec FX (Becton-Dickinson, San Jose, CA) were plated onto specific agar then all grown colonies were identified using MALDI Biotyper sirius IVD System (Bruker Daltonics GmbH & Co. KG Life Sciences, Bremen, Germany) or with Vitek galleries by Biomérieux (Craponne, France). This process included the search for

Typhoid fever, pneumococcal disease, or E. coli bacteremia, with Antibiotic Susceptibility Testing when needed.

**Sepsis:** Organ dysfunction resulting from infection during pregnancy, as defined by WHO [16].

The retained cause of the infection was based on synthesizing the case records and reaching consensus among the multidisciplinary maternity ward staff.

## Antenatal care

Antenatal care was provided by obstetricians and midwives in accordance with the recommendations for clinical practice by the French College of Gynecologists and Obstetricians. All patients were clinically evaluated. Fetal wellbeing was confirmed, if necessary, by cardiotocography. Ultrasonographic evaluation was performed depending on the type of suspected infection and repeated as necessary.

Hospitalization depended on both clinical presentation and social conditions (living far from hospital, lack of or poor pregnancy follow-up, lack of health insurance coverage, etc.).

## Objectives

**Primary objective.** Identify the cause(s) of fever during pregnancy in a mixed multiethnic population living along the Maroni River bordering French Guiana and Suriname.

**Secondary objectives.** Reporting of all maternal complications directly related or unrelated to febrile infection. To illustrate the impacts of fever on neonatal and maternal health (e.g., stillbirth, preterm delivery).

## Data collection

Data were collected retrospectively and anonymously from the delivery registry at Saint Laurent du Maroni Hospital. The midwife routinely entered pregnancy outcomes and complications for all women who had delivered at the West French Guiana hospital into a computerized database. We included all entries dated between January 1, 2013, and December 31, 2021. The following information was obtained from the maternal charts: demographics, medical history, prenatal care, ultrasound scans, delivery parameters, biological testing, and pathological examination of the fetus and placenta. Data on febrile women were cross-referenced with pregnancy outcome as recorded in the delivery registry. If women had more than one pregnancy during the study period, each pregnancy was counted separately.

## Statistical analysis

Data were anonymized and analyzed using STATA v18.0. Continuous variables are given as median and interquartile range (25% vs 75%). The comparison of proportions was performed using a Chi-squared test or Fisher's exact test, as appropriate. The unpaired Student's t-test and Mann–Whitney U-test were used to compare groups of continuous normally and non-normally distributed variables, respectively. The odds ratios (OR) of birth by caesarean sections, prematurity (<34 weeks of gestation), and stillbirth were adjusted (aOR) for potential confounders between the groups with and without documented fever. Presented p-values are two-sided, and the significance level was set to 5% (p = 0.05) for all statistical tests.

A descriptive analysis of the causes of fever in the sub-group having fever outside labor and delivery was presented.

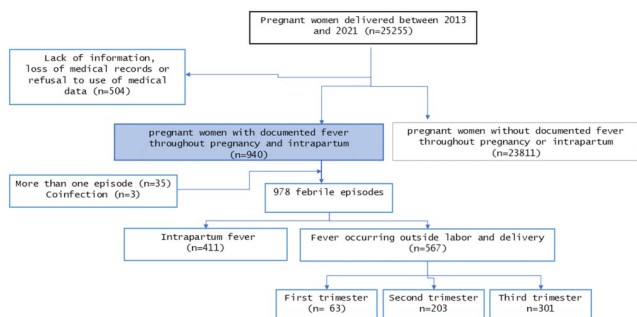

**Fig 1. Flow diagram of patient recruitment in this retrospective study.** Second trimester: 14 weeks and 0 days to 27 weeks and 6 days.

## Results

### Study population

During the 9-year study period, a total of 940 pregnant women with at least one episode of fever during their pregnancy were included. Among them, 11 had twins (11/940, 1.1%). In total, 978 febrile infections were analyzed. Obstetrical outcomes were compared to 23,811 deliveries that occurred during the same period without documented fever (Fig 1).

Mothers had one febrile episode in 96% (902/940), two or more episodes in 3.7% (35/940), and a coinfection at the time of diagnosis in 0.31% (3/940) of cases. Women who delivered during the same period without febrile episodes served as controls. The demographic characteristics of the women with fever are described in Table 1. In total, the maternal mean age at delivery was 26.8 years (interquartile range 21–32 years). After excluding fever during labor, infection occurred in 11% (63/567), 35.8% (63/567), and 53% (301/567) of women in their first, second, and third trimester of pregnancy, respectively. Table 2 presents a comparison of febrile and non-febrile pregnant women in terms of maternal risk factors for fever, including their medical condition before pregnancy and obstetric history. The mean age of women in the fever group was significatively lower than in the control group. There were also significantly more nulliparous, and women aged under 20 years in the febrile group. Women with fever were more likely to have lower gravida and parity.

There were also significant differences between women with and without documented fever in homozygous sickle cell disease and gestational diabetes.

We observed no differences in terms of body mass index (BMI), BMI >30 kg/m2, scarred uterus, or other medical conditions before and during pregnancy.

**Main causes of fever.** The median gestational age at fever onset pre-partum (n = 567) was 27 weeks. For women with febrile episodes occurring while in labor (n = 411), mean onset was at 37 weeks. Among the 978 episodes of infection, the causes were unknown in 7.6% (75/978) of cases and uncertain in 0.9% (9/978) of cases. When considering only confirmed causes of febrile episodes occurring outside labor and delivery (n = 483), almost one third were related to arbovirus infection (146/483,30.2%). Arbovirus infections included CHIKV (84/146, 57.5%), dengue virus (36/146, 24.6%), ZIKV (23/146, 15.7%), and others (Mayaro, Tonate virus, acute flavivirus) (3/146, 2%). Other causes included urinary tract infections (134/483,27.7%), chickenpox (27/483, 5.6%), gastrointestinal (14/483, 2.9%) and pulmonary infections (10/483, 2%) (Table 3). Gastrointestinal infections were dominated by shigellosis (9/14, 64%). One case of confirmed fever was due to *Listeria monocytogenes*.

**Table 2. Comparison of maternal baseline characteristics and pregnancy co-morbidities between pregnant women with and without documented fever.** Post-partum fever has been excluded from the analysis. IQR, interquartile range; BMI, body mass index.

| | Pregnant women with documented fever (n = 940) | Pregnant women without documented fever (n = 23811) | p-value |
|---|---|---|---|
| Maternal age, median (IQR) | 25.8 (20, 31) | 26.8 (21, 32) | <**0.001** |
| Age | | | |
| ≤19 | 189 (20%) | 4190 (17.6%) | 0.002 |
| 20–29 | 471(50%) | 11195 (47%) | |
| 30–39 | 246 (26.1%) | 7321 (30.7%) | |
| 40+ | 30 (3.1%) | 1075 (4.5%) | |
| BMI, median (IQR), kg/m2 | 27.5 (22.4–31) | 27.1 (22–31) | 0.57 |
| Gravidity, median (IQR) | 4.1 (1–6) | 4.5 (2–6) | <**0.001** |
| Parity, median (IQR) | 2.1 (0–4) | 2.8 (1–4) | <**0.001** |
| Nulliparous | 315 (33.5%) | 5461 (22.9%) | <**0.001** |
| Type of gestation | | | |
| Single | 929 (98.8%) | 23481(98.6%) | 0.6 |
| Twins | 11(1.2%) | 326 (1.3%) | |
| More | 0 (0%) | 4 (0.1%) | |
| Scarred uterus | 137 (14.5%) | 3164 (13.3%) | 0.2 |
| **Medical conditions before pregnancy** | | | |
| Chronic high blood pressure | 19 (2%) | 394 (1.7%) | 0.3 |
| Pre-existing diabetes mellitus | 8 (0.8%) | 138 (0.6%) | 0.8* |
| Homozygous sickle cell disease | 10 (1.06%) | 135 (0.6%) | **0.05** |
| Human immunodeficiency virus | 7 (0.74%) | 281 (1.18%) | **0.2*** |
| Cardiac diseases | **0 (0%)** | **13 (0.05%)** | **1*** |
| Renal diseases* | **2 (0.2%)** | **21 (0.09%)** | **0.2*** |
| **Co-morbidities during pregnancy** | | | |
| Severe anemias < 8g | **73 (7.7%)** | **1532 (6.4%)** | **0.1** |
| Preeclampsia/hypertension | 83 (8.8%) | 1902 (8%) | 0.3 |
| Gestational diabetes | 51 (5.4%) | 853 (3.6%) | **0.003** |

* Fisher's exact test

Discrepancies between the denominator for some of the categories and the number of pregnancies included in the study are due to missing data

Among febrile episodes occurring outside labor and delivery, we have data on hospitalization available for 509 patients. Of these, 53% (270/509) required hospitalization, with a median stay of 4 days (range: 1 to 33 days). Empiric antibiotics were administered in 38.1% (194/509) of cases.

Maternal febrile infection was complicated by sepsis in 2.8% (27/940), admission to intensive care in 2.7% (26/940), and maternal death in 0.1% (1/940) of cases. Cesarean section was more common among women with febrile episodes compared to controls, even when considering parity, age, and the presence of gestational diabetes or sickle cell disease as potential confounders (19.8% vs 15.5%, aOR 1.3 [95% CI 1.14–1.6], p < 0.0001, Table 4). Women with fever had a greater risk of stillbirth (5.5% versus 1.92%, aOR 2.7 [95% CI 2–3.7], p < 0.0001) and preterm delivery before 34 weeks of gestation (7.2% vs 4.7%, aOR 1.5 [95% CI 1.2–2], p = 0.001) than women without fever during their pregnancy, even when considering potential confounders.

Fever was considered as preventable in 27 cases of varicella. Vaccination rates for influenza and COVID-19 (80 cases) febrile episodes were not available in the medical charts.

**Table 3. Confirmed causes of fever among pregnant women in Western French Guiana and along the Maroni River.**

| Causes of fever | Number n = 483 | Percentage (%) |
|---|---|---|
| **Arboviruses** | 146 | 30.2 |
| CHIKV | 84 | 17.4 |
| DENV | 36 | 7.4 |
| ZIKV | 23 | 4.7 |
| Other arboviruses | 3 | 0.6 |
| **Urinary tract infections** | 134 | 27.7 |
| COVID 19 | 64 | 13.2 |
| Chickenpox | 27 | 5.5 |
| Influenza | 16 | 3.3 |
| Gastrointestinal infections | 14 | 2.9 |
| Shigellosis | 9 | 1.8 |
| Viral | 4 | 0.8 |
| Salmonellosis | 2 | 0.4 |
| Ear, nose, and throat disorders | 12 | 2.4 |
| Skin lesions and abscesses | 10 | 2 |
| Pulmonary infections | 10 | 2 |
| Leptospirosis | 9 | 1.8 |
| Malaria | 8 | 1.6 |
| STI | 7 | 1.4 |
| Secondary syphilis | 4 | 0.8 |
| HSV primary genital infection | 3 | 0.6 |
| *Coxiella burnetii* (Q fever) | 5 | 1 |
| Others | 21 | 4.3 |
| **Total** | 483 | 100 |

This table only lists cases having occurred outside labor and delivery (n = 483). STI, sexually transmitted infections; CHIKV, Chikungunya virus; DENV, dengue virus. ZIKV, Zika virus.

Other arboviruses: Tonate virus, Mayaro, acute flavivirus without precision

Others: Meningoencephalitis, Coxsackievirus, *Listeria monocytogenes*, myoma necrobiosis, Epstein-Barr virus, toxoplasmosis, septicemia of undetermined origin, uterine rupture, cytomegalovirus, tuberculosis.

**Table 4. Comparison of main perinatal outcomes between pregnant women with documented fever and pregnant women without documented fever.**

| | Pregnant women with documented fever (n = 940) | Pregnant women without documented fever (n = 23811) | aOR* [95%CI] | P value |
|---|---|---|---|---|
| **Stillbirth** | 52/940 (5.5%) | 459/23811 (1.92%) | 2.7 [2–3.7] | 0.001 |
| **Cesarean delivery** ** | 180/909 (19.8%) | 3690/23803 (15.5%) | 1.3 [1.1–1.6] | 0.001 |
| **Preterm birth <34wg** | 68/940 (7.2%) | 1119/23811 (4.7%) | 1.5 [1.2–2] | 0.001 |
| **Preterm birth <37wg** | 247/940 (26.3%) | 5738/23811 (24.1%) | 1.1 [0.9–1.3] | 0.12 |

* Odds Ratios were adjusted (aOR) on the parity, the age, gestational diabetes, and sickle cell diseases

** Discrepancies between the denominator for some of the categories and the number of pregnancies included in the study are due to missing data.

wg, weeks of gestation

## Discussion

### Principal findings of the study

Although fever during pregnancy is a common reason for women to visit emergency departments, reliable numbers on this topic are limited. In this study, we found that at least 3.7% of pregnant women had experienced confirmed fever during pregnancy and labor resulting in severe adverse events. The most common causes of fever were arboviruses and urinary tract infection. Almost one third of febrile pregnant women had a diagnosis of arbovirus infection during their pregnancy. *Listeria monocytogenes* was uncommon. Nulliparity and younger age were the main risk factors. We suggest that the increased susceptibility among young parturient individuals may be attributed to their potential exposure to arboviruses, which were the primary cause of fever in our study, in addition to sexually transmitted infections. Regarding arboviral infections (such as DENV, CHIKV, and ZIKV) and chickenpox, prior exposure could potentially confer immunity, contributing to the variance in susceptibility observed. Given the global context of emerging and re-emerging infectious agents and the lack of scientific data specific to pregnant women, our findings could help to inform patients and healthcare providers, facilitating the development of appropriate management protocols for febrile pregnant women in these geographical settings and for travelers having visited exposed areas.

### Main strengths and limitations of the study

To the best of our knowledge, this is the first large description of the extent of fever during pregnancy in the Amazonian basin and of its main causes and consequences. This study included a large cohort of 940 patients over a long 9-year period that covered different arbovirus outbreaks. The West French Guiana Hospital is the only reference hospital for a relatively large area and therefore allows representative descriptions of the population. We used a standardized protocol for the management of fever during pregnancy based on both clinical and biological assessments. The clinical evaluation was performed by senior medical doctors with experience in tropical diseases.

Our study does not provide information on the frequency of each infection during pregnancy but rather focuses exclusively on infections accompanied by documented fever. We intentionally refrained from detailing the specific consequences of each infection as we wanted to narrow the scope of our investigation to infections manifesting with documented fever in a hospital setting. In fact, many infections can cause fetal abnormalities and pregnancy adverse outcomes without being symptomatic such as ZIKV, syphilis, and some arboviruses [17,18]. Hence, asymptomatic pregnant women can still experience such adverse outcomes.

This study had several limitations. Firstly, we cannot ensure that the control group never experienced any fever (undocumented). Although patients were advised to visit the emergency department in case of fever, some patients living in isolated areas with limited or no access to transportation may only reach the hospital after their fever has resolved. Additionally, some women may have consulted their primary physician first. In practice, all consultations during pregnancy, even those outside the hospital, are recorded in a pregnancy booklet and in the final pregnancy report. Secondly, we only considered cases of documented fever to avoid overestimating the study population. Patients who self-reported a fever without possessing documented evidence were not categorized as part of the fever group.

Thirdly, our population is very heterogeneous: 70% are of African origin, and there are significant Amerindian, Asian, and European communities [19]. Unfortunately, data such as ethnicity, socioeconomic level, lifestyle, and educational level were not available for this study. Social precariousness, malnutrition, and co-morbidities during pregnancy are particularly

frequent in western French Guiana and could increase the risks of infection and of adverse outcomes; they should therefore be considered as potentiating factors [20]. Alcohol, tobacco, and drug use may be underreported by patients.

Finally, the retrospective collection of information from patient charts was sometimes incomplete which may have biased the analysis.

## Comparison with existing literature and implications

Fever can reveal an infection which can lead to maternal and fetal adverse outcomes with life-threatening complications [7,21]. Infection causes maternal and fetal/neonatal morbidity and/or mortality and can lead to a wide range of obstetrical complications. Arboviruses and malaria are associated with adverse pregnancy outcomes like preterm birth, low birthweight, and stillbirth [22]. The threat presented by these infections to both the mother and the developing fetus can stem from various mechanisms including teratogenic effects, placental infection, and maternal complication [7,23,24]. Hyperthermia can cause a twofold increase in uterine activity [25]. Early diagnosis and rapid treatment are crucial to prevent complication. A significant decrease in uterine activity was observed after antibiotic administration in women with pyelonephritis [23]. Investigating the incidence and underlying causes of fever during conception is of paramount importance. There is substantial evidence to suggest that maternal fever during pregnancy, irrespective of the pathogen involved, may represent a risk factor for neurodevelopmental disorders in the offspring [8,9], for autism spectrum disorders [10], and for cardiovascular malformations [26]. However, not all studies have found these increased risks [27,28]. Furthermore, there is a relationship between the severity of intrapartum fever and neonatal outcomes [29]. A distinction must be made between fever during and outside labor. Intrapartum fever is common: 6.8% [30], often with specific causes. The prevalence has increased in recent decades along with the increase in use of neuraxial anesthesia [31].

Some populations are particularly at risk of infection in the Amazonian context. For instance, illegal gold miners are exposed to several specific illnesses (HIV, leishmaniasis arboviruses and digestive disorders) [32]. Although these precarious populations are more exposed [33], infections are certainly under-diagnosed and under-reported in this geographic area. In fact, the Amazon Basin has been affected by many vector-borne and zoonotic outbreaks [3,4,34]. The multicultural and polyethnic populations along the Maroni River, representing the border between Suriname and French Guiana, are often mobile, switching their place of residence between the Suriname to the French Guiana sides of the river. In this context, outpatient management of fever during pregnancy becomes complicated, and results should be interpreted with caution, especially in isolated and precariousness populations. The patient history and the results of their clinical examination must be considered as well as biological testing. Despite extensive testing, the cause of fever can remain elusive. The prevalence of unexplained fever may vary depending on diagnostic procedures. In our study population, the cause of fever remained unknown in 8.5% of pregnant women, while in mainland France (S1 Table), this percentage can reach up to 15% [35].

Biological diagnosis of infections during pregnancy requires several considerations. For example, for a molecular diagnosis of arboviruses, it is crucial to consider the short window of viral secretion compared to other infections such as cytomegalovirus [34,36]. The presence of fever allows the infection onset to be determined with regard to pregnancy term, a critical factor in assessing its impact on the course of pregnancy. Women in their first trimester are particularly at risk [7,36]. Therefore, it is necessary to adjust the monitoring protocols to both the underlying pathogen and the timing of infection during pregnancy.

### Research implication

Both maternal and fetal adverse outcomes are associated with infections during pregnancy [7,21]. Our study, like many others, has demonstrated that infection remains a life-threatening condition for both the pregnant woman and the fetus [37,38]. Particular attention must be paid to malaria and arboviruses which can be particularly grave during pregnancy. Rapid diagnosis of bacterial and viral infections can help reduce unnecessary use of antibiotics during pregnancy and allow appropriate care including outpatient management. Careful monitoring of febrile pregnancies is crucial. It is highly advisable to provide all relevant information to everyone involved in antenatal care, particularly when conducting ultrasound scans following arboviral infections.

Close clinical and biological monitoring, as well as appropriate treatment, are essential for pregnant individuals experiencing fever to ensure tailored care and reduce maternal infectious morbidity. The approach should be adapted based on the specific infectious agent involved. Furthermore, it is crucial to consider the possibility of mother-to-infant transmission when dealing with certain infectious pathogens such as arboviruses and syphilis infection. Pregnant women living or visiting the Amazon rainforest should be advised to take long clothing, use mosquito repellent lotions suitable for pregnancy, and to sleep under a mosquito net at night. After visiting the Amazon Basin, pregnant travelers who develop a fever are strongly advised to seek immediate antenatal care and to inform their healthcare provider.

### Conclusion

This study summarized various causes of fever in a general tropical and specifically Amazonian context. The study period covered a sufficiently long period that spanned several epidemics in French Guiana such as DENV, CHIKV, ZIKV, influenza, and COVID-19. Our results indicated that fever management protocols during pregnancy must be adapted to the specific context of these regions. Interpretation of fever should consider the epidemiological context. These infections adversely affect pregnancy in these areas. It is important to ensure that febrile pregnant women with no evident cause are screened as early as possible for arboviruses, malaria, and sexually transmitted diseases. If positive, patients should be referred for maternal and fetal assessment. This study can help clinicians manage fever in pregnant travelers and in their partners after visiting exposed areas.

### Supporting information

**S1 Table. Comparative analysis of fever etiologies during pregnancy in west French Guiana and mainland France.**
(DOCX)

### Author Contributions

**Conceptualization:** Najeh Hcini, Véronique Lambert, Olivier Picone, Jean-Francois Carod, Mérédith Mathieu, Ferroudja Akli, Gabriel Carles, Célia Basurko, Léo Pomar, Loïc Epelboin, Mathieu Nacher.

**Data curation:** Najeh Hcini, Véronique Lambert, Mérédith Mathieu, Romane Cousin, Ferroudja Akli, Gabriel Carles.

**Formal analysis:** Najeh Hcini, Mérédith Mathieu, Romane Cousin, Gabriel Carles, Léo Pomar, Loïc Epelboin, Mathieu Nacher.

**Investigation:** Najeh Hcini, Jean-Francois Carod, Romane Cousin, Gabriel Carles, Célia Basurko.

**Methodology:** Najeh Hcini, Véronique Lambert, Mérédith Mathieu, Ferroudja Akli, Gabriel Carles, Célia Basurko, Loïc Epelboin, Mathieu Nacher.

**Project administration:** Gabriel Carles, Mathieu Nacher.

**Software:** Léo Pomar.

**Supervision:** Olivier Picone, Jean-Francois Carod, Gabriel Carles, Célia Basurko, Loïc Epelboin, Mathieu Nacher.

**Validation:** Najeh Hcini, Véronique Lambert, Olivier Picone, Jean-Francois Carod, Gabriel Carles, Célia Basurko, Loïc Epelboin, Mathieu Nacher.

**Visualization:** Léo Pomar.

**Writing – original draft:** Najeh Hcini, Véronique Lambert, Olivier Picone, Jean-Francois Carod, Mérédith Mathieu, Romane Cousin, Ferroudja Akli, Gabriel Carles, Célia Basurko, Léo Pomar, Loïc Epelboin, Mathieu Nacher.

**Writing – review & editing:** Najeh Hcini, Véronique Lambert, Mérédith Mathieu, Gabriel Carles, Léo Pomar.

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
