## [Decision Letter · Decision Letter 0]

22 Aug 2023

Dear MD Hcini,

Thank you very much for submitting your manuscript "Causes and consequences of fever in Amazonian pregnant women: A large retrospective study from French Guiana" for consideration at PLOS Neglected Tropical Diseases. As with all papers reviewed by the journal, your manuscript was reviewed by members of the editorial board and by several independent reviewers. In light of the reviews (below this email), we would like to invite the resubmission of a significantly-revised version that takes into account the reviewers' comments. 

Your manscript was considered of value and should be of interest to many people but both reviewers had some significant comments for items that should be clarified and suggestions for improvement. Please review these comments carefully, modify your manuscript accordingly and resubmit along with point-by-point comments concerning the changes you have made or justification for not changing.

We cannot make any decision about publication until we have seen the revised manuscript and your response to the reviewers' comments. Your revised manuscript is also likely to be sent to reviewers for further evaluation.

Sincerely,

Richard A. Bowen

Academic Editor

Andrea Marzi

Section Editor

Your manscript was considered of value and should be of interest to many people but both reviewers had some significant comments for items that should be clarified and suggestions for improvement. Please review these comments carefully, modify your manuscript accordingly and resubmit along with point-by-point comments concerning the changes you have made or justification for not changing.

Reviewer's Responses to Questions

**Key Review Criteria Required for Acceptance?**

**Methods**

-Are the objectives of the study clearly articulated with a clear testable hypothesis stated?

-Is the study design appropriate to address the stated objectives?

-Is the population clearly described and appropriate for the hypothesis being tested?

-Is the sample size sufficient to ensure adequate power to address the hypothesis being tested?

-Were correct statistical analysis used to support conclusions?

-Are there concerns about ethical or regulatory requirements being met?

Reviewer #1: (No Response)

Reviewer #2: The authors have presented an extended retrospective cohort study on the causes of fever in pregnancy in an equatorial setting in South America. The study extends over nine years and thus should represent a realistic picture of the major risk factors of fever in pregnancy for the area.

There are a few points for consideration prior to publication. 

Introduction

In the first sentence of the introduction, lines 67 = 68, the authors should include fetal morbidity and mortality as a risk factor of infection during pregnancy.

Lines 77 – 79: “Although being the origin an endemic area of several epidemics such as arboviruses, there are few information’s about in fever during pregnancy in the Amazon basin” needs to be revised to make English sense.

Line 83: Please change “Scientific data supports” to “Scientific data support” – “data” is the plural form of “datum.”

Methods

Lines 103 – 114: This whole passage needs to be revised to make sense in English.

Line 103 – 105: “…documented pregnancies with an auricular temperature taken with an ear thermometer temperature with an infrared emission detection greater than or equal to 38 °C ear temperature with an infrared emission detection thermometer were included” – repetitive information should be excluded.

Line 107 – 109: The exclusion criteria can be included in the flow chart of how subjects were selected for exclusion – see the literature for examples in similar studies.

Line 106: Correct “Monocytogenes” to lower case, and italicise elsewhere in the manuscript as well>

The heading “protocol description” should be changed to diagnosis. For the laboratory diagnosis include the manufacturer of the test, city and country of origin, or if this was developed in-house, reference the methodology/provide primer sequences where relevant. What blood culture system was used? Define PAMG-1, EBV, CMV and CNGOF– although these are well-recognised acronyms, they are not all exclusive.

Line 171: Correct the spelling of Listeria monocytogenes and italicise.

Were other bacterial causes of respiratory infection e.g. Haemophilus influenzae other than Hib, tuberculosis, Mycoplasma pneumoniae, or bacterial causes of diarrhoea and dysentery that may be associated with fever, including Salmonella, Shigella or Campylobacter sought, and if so, how? Did blood cultures include examination for typhoid fever, pneumococcal disease or E. coli bacteraemia (especially secondary to pyelonephritis)? What about leptospirosis, Hepatitis A, malaria and yellow fever?

Lines 201 – 202: “(living far from hospital, lack or poor pregnancy follow-up, lack of health insurance coverage …) needs to be rephrased to clarify sense.

Line 258: “Midwives” or “the midwife”

Table 1: Correct spelling of Coxiella burnetii and Listeria monocytogenes and italicise.

Did the authors successfully get permission from both the cases and the control group to use the data?

**Results**

-Does the analysis presented match the analysis plan?

-Are the results clearly and completely presented?

-Are the figures (Tables, Images) of sufficient quality for clarity?

Reviewer #1: (No Response)

Reviewer #2: Results

Line 282: What is meant by single deliveries – do the authors mean live births, as per the table some of these were multiple births?

Line 283: D the authors mean “same period”?

Figure 1: Given the numbers, the study design to select the study participants appears suitable, but may be easier to grasp if the authors could include in the flow chart how many pregnant women presented during the study period in total, and how many were excluded because of refusal to sign consent, insufficient data etc.

Line 295: “Gravidity” may be better than “gravida.”

Table 2: Under “Parity” correct “2.8(1-, 4)” – what is the actual range? Correct spelling of “burnetiid” - a burnetiid is an extinct crocodilian reptile and tis error may be a function of the computer’s autocorrect.

Line 356: Do the authors mean “empiric antibiotics?”

**Conclusions**

-Are the conclusions supported by the data presented?

-Are the limitations of analysis clearly described?

-Do the authors discuss how these data can be helpful to advance our understanding of the topic under study?

-Is public health relevance addressed?

Reviewer #1: (No Response)

Reviewer #2: Discussion and conclusion

Lines 453 – 454: Fever “was 8.5% in our study population versus 15% in 100 pregnant women of metropolitan France” – can the authors elaborate on this? Is this because they missed cases, had a more stringent definition, or fever is less common in this [population because childhood exposures to common pathogens has increased the population immunity on the area?

Lines 454 – 462: The authors seem to have conflated a number of ideas in this paragraph, confusing the point they are trying to make. This needs revision.

The authors have presented a fairly critical view of their study and placed the results in context, regarding the socioeconomic challenges etc. I think it would be important to highlight that their results are very specific to the area, given the increased prevalence of certain regional conditions such as Zika virus. Generally, I think there is scope to expound more on the causes of fever in pregnancy in the area, possibly in a supplementary table if there is no room in the text, as it would then be easier to compare these results with similar studies elsewhere.

**Editorial and Data Presentation Modifications?**

Reviewer #1: (No Response)

Reviewer #2: See above

**Summary and General Comments**

Reviewer #1: (No Response)

Reviewer #2: It would also be advantageous to have the manuscript reviewed by a native English speaker, as some passages are difficult to understand.

PLOS authors have the option to publish the peer review history of their article (what does this mean?). If published, this will include your full peer review and any attached files.

Reviewer #1: Yes: Surendran Deepanjali

Reviewer #2: No
---

## [Editor Report · Decision Letter 1]

12 Oct 2023

Dear MD Hcini,

We are pleased to inform you that your manuscript 'Causes and consequences of fever in Amazonian pregnant women: A large retrospective study from French Guiana' has been provisionally accepted for publication in PLOS Neglected Tropical Diseases.

Best regards,

Richard A. Bowen

Academic Editor

Andrea Marzi

Section Editor

Thank you for your careful and comprehensive editing in response to the critiques and suggestions from reviewers. I believe that the manuscript has been improved significant, especially in clarity, and is acceptable for publication.

---

## [Editor Report · Acceptance letter]

20 Oct 2023

Dear MD Hcini,

We are delighted to inform you that your manuscript, "Causes and consequences of fever in Amazonian pregnant women: A large retrospective study from French Guiana," has been formally accepted for publication in PLOS Neglected Tropical Diseases.

Best regards,

Shaden Kamhawi

co-Editor-in-Chief

Paul Brindley

co-Editor-in-Chief
